# Identifying risk factors for cancer-specific early death in patients with advanced endometrial cancer: A preliminary predictive model based on SEER data

**Jing Yang** [ID]*, **Qi Tian, Guang Li, Qiao Liu, Yi Tang, Dan Jiang, Chuqiang Shu**

Department of Obstetrics and Gynecology, Hunan Provincial Maternal and Child Health Care Hospital, Changsha, Hunan, P.R. China

* yjing2274@gmail.com

## Abstract

### Objective

To identify risk factors associated with cancer-specific early death in patients with advanced endometrial cancer and to develop a preliminary nomogram prediction model based on these factors, with an emphasis on the potential implications for clinical practice.

### Methods

Patients from the Surveillance, Epidemiology, and End Results (SEER) database in the United States from 2018 to 2021 were included in the study. The study data was randomly divided into a training cohort and a validation cohort at a ratio of 7:3. Multivariate logistic regression analysis was performed in the training cohort to screen for risk factors for cancer-specific early mortality in advanced endometrial cancer patients, and a preliminary nomogram prediction model was further constructed. The results of the Receiver Operating Characteristic (ROC) curve, calibration analysis, and clinical decision curve analysis (DCA) were presented for transparency.

### Results

Significant risk factors for cancer-specific early death were identified, including tumor size (≥101 mm, OR = 2.11, $P < 0.001$), non-endometrioid histology (OR = 3.11, $P < 0.001$), high tumor grade (G3, OR = 2.68, $P = 0.007$), advanced tumor stages (T3-T4, OR = 1.84, $P = 0.004$), and metastatic stage (M1, OR = 2.05, $P < 0.001$), as well as the presence of liver metastases (OR = 2.21, $P = 0.005$) and brain metastases (OR = 8.08, $P < 0.001$). Protective factors that were significantly associated with a reduced risk of early death included hysterectomy (OR = 0.13, $P = 0.012$), radical surgery (OR = 0.21, $P < 0.001$), radiation therapy (OR = 0.40, $P < 0.001$), and chemotherapy (OR = 0.31, $P < 0.001$). A preliminary nomogram model was demonstrated adequate predictive performance with AUC values of 0.89 (95% CI 0.87 to 0.91) in the training cohort and 0.88 (95% CI 0.84 to 0.91) in the

**Data availability statement:** This study analyzed publicly available datasets, which can be accessed at https://seer.cancer.gov/data/. Additionally, we have provided a minimal dataset necessary to replicate the findings of our study, which can be accessed via the following DOI: https://doi.org/10.6019/S-BSST1710

**Funding:** This study was supported by Hunan Province Science and Technology Innovation Platform and Talents Program—Hunan Province Cervical Cancer Prevention and Clinical Research Center Fund (Approval Number: 2021SK4021). Natural Science Foundation of Hunan Province Department Joint Fund Project (2023JJ60013). Hunan Provincial Natural Science Foundation Medical and Health Industry Joint Fund (NO. 2024JJ9334).

**Competing interests:** The authors have declared that no competing interests exist.

validation cohort. The model's predictive performance was further supported by the calibration and DCA analyses, suggesting its potential clinical utility.

## Conclusion

This study identified key risk factors for early cancer-specific mortality in patients with advanced endometrial cancer. The preliminary nomogram model holds promise for predicting early death risk and could be valuable in clinical practice. Future work may explore its performance with additional data to ensure broad applicability.

## Introduction

The incidence of endometrial cancer (EC) is increasing worldwide, with approximately 10% to 15% of patients diagnosed at an advanced stage. The 5-year survival rate for patients with stage III and IV is about 48% and 15% respectively [1]. In endometrial cancer, FIGO (International Federation of Gynecology and Obstetrics) staging and TNM (Tumor, Node, Metastasis) staging are commonly used to predict patient survival and guide treatment plans. However, these staging systems primarily focus on tumor characteristics and do not account for other important prognostic factors, limiting their predictive power. Identifying risk factors for cancer-specific early death is crucial for enhancing clinical management strategies and informing personalized treatment approaches in advanced endometrial cancer. Prognostic factors for endometrial cancer include age, tumor pathological type, pathological grading, depth of myometrial invasion, lymph node metastasis, and treatment modality, etc. The prognosis for patients at the same stage can vary significantly due to these factors. It should be highlighted that the current literature does not offer comprehensive predictive models specifically designed for patients with advanced endometrial cancer. While molecular staging has been shown to predict prognosis and guide effective clinical treatments accurately, it is not widely used in practice due to the complexity and high cost of the testing process. Additionally, molecular staging has limited value in advanced endometrial cancers. Therefore, clinical staging alone is insufficient for predicting the prognosis of patients with advanced endometrial cancer. In light of this, we conducted a study to analyze the risk factors for cancer-specific death in patients with advanced endometrial cancer and to construct a preliminary nomogram prediction model for cancer-specific early death. This model serves as an initial step towards a more refined tool that may aid clinicians in risk stratification and decision-making.

The Surveillance, Epidemiology, and End Results (SEER) database is one of the largest public cancer databases in the U.S. The database has a large sample size, complete follow-up information, and contains a large amount of diagnostic and survivorship data for a wide range of cancer cases, and it has been used in a wide variety of clinical studies for a wide range of tumors. It has been widely used in clinical research on a variety of tumors. In this study, we collected demographic data, clinicopathological data, and follow-up data of patients with stage III-IV endometrial cancer from the SEER database, analyzed the data to explore the risk factors for cancer-specific early death in patients with stage III-IV endometrial cancer. Based on the identified risk factors, we have constructed a preliminary prediction model for the cancer-specific early death of stage III-IV endometrial cancer patients. This initial model is designed to lay the groundwork for future refinements and serves as a starting point for developing more sophisticated tools. By visualizing the model through a nomogram, we aim to harness its potential to assist clinicians in the early identification of high-risk patients, thus paving the way for the formulation of personalized treatment plans.

## Data and methods

### Ethics statement

The SEER database does not require informed patient consent as cancer is a reportable disease across all states in the United States. This study aligns with the ethical guidelines set forth in the 1964 Helsinki Declaration and its subsequent amendments or equivalent ethical standards.

### Sources of case data

Patient data on stage III-IV endometrial cancer during 2018-2021 were obtained by downloading from the U.S. SEER database (version: SEER*Stat8.4.3), The diagnosis of stage III-IV endometrial cancer collected and downloaded uses the AJCC 7th edition TNM staging and is converted to FIGO (2009) staging. The case screening and study flow are illustrated in (Fig 1).

### Inclusion and exclusion criteria

Inclusion criteria: The inclusion site codes were C54.0 (Endometrial carcinoma of the isthmus uteri), C54.1 (Endometrial carcinoma of the corpus uteri), C54.2 (Endometrial carcinoma of the fundus uteri), C54.3 (Endometrial carcinoma of the overlapping sites of the corpus uteri), C54.8 (Endometrial carcinoma of the overlapping sites of the uterus), C54.9 (Endometrial carcinoma of the uterus, unspecified), and C55.9 (Malignant neoplasm of the uterus, unspecified). The histological codes were 8380/3 (Endometrioid adenocarcinoma), 8441/3 (Serous cystadenocarcinoma), 8480/3 (Mucinous adenocarcinoma), 8020/3 (Undifferentiated carcinoma), and 8930/3 (Adenosarcoma), according to the International Classification of Tumor Diseases, Third Edition (ICD-O-3).

Exclusion criteria: TNM stage I-II; Unknown marital status, ethnicity, age, or basis of diagnosis; Unknown tumor size, histological classification, histology grade, surgery, or T, N, and M stage.

### Data collection

Basic Data: Age (categorized using x-tile software to determine optimal cut-off values for age groups: ≤64 years, 65-74 years, and ≥75 years) (Fig 2), marital status (Married: individuals in legal or common-law marriages.

Single: individuals who are unmarried, divorced, or widowed), and ethnicity. Clinical and Pathological Data: Tumor diameter (x-tile software used to determine optimal cut-off values for diameter groups: ≤55mm, 56-100mm, and ≥101mm) (Fig 3), histological classification, histology grade, T-stage, N-stage, M-stage, surgical method for primary site, regional lymph nodes removed, radiotherapy status, chemotherapy status, and systemic therapy status, tumor metastasis (including brain metastasis, liver metastasis, lung metastasis, and bone metastasis). Survival Data: Survival time, survival status, and cause of death. The primary outcome of this study was cancer-specific early death, defined as death caused by endometrial cancer within 6 months of the initial diagnosis [2–5].

### Statistical methods

SPSS 24.0 and R 4.4.0 software were used for statistical analysis. The study data were randomly divided into a training cohort and a validation cohort in a 7:3 ratio. Comparisons between factors in the training and validation cohorts were performed using chi-square tests for RxC contingency tables. The primary outcome variable for the logistic regression analysis was defined as 'cancer-specific early death,' which refers to death attributable to endometrial cancer within six months following the initial pathological diagnosis. The training cohort was

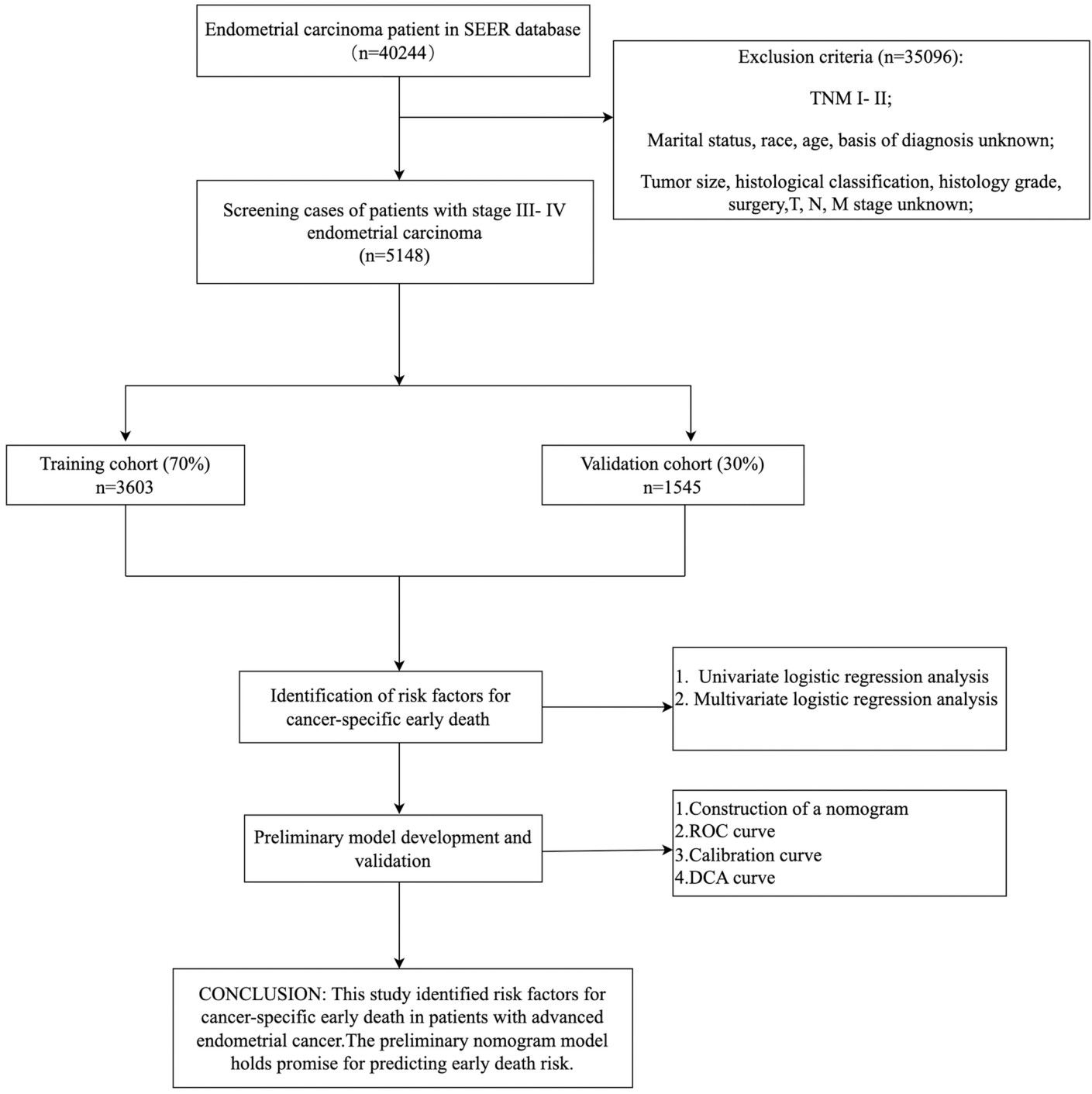

**Fig 1. The patient selection procedure and the process involved in this research.**

used to analyze the factors affecting cancer-specific early death in patients with stage III-IV endometrial cancer and to establish a preliminary nomogram model, while the validation cohort was used to verify the nomogram model. Univariate logistic regression analysis was performed in the training cohort, and variables with $P < 0.05$ were included in the multivariate

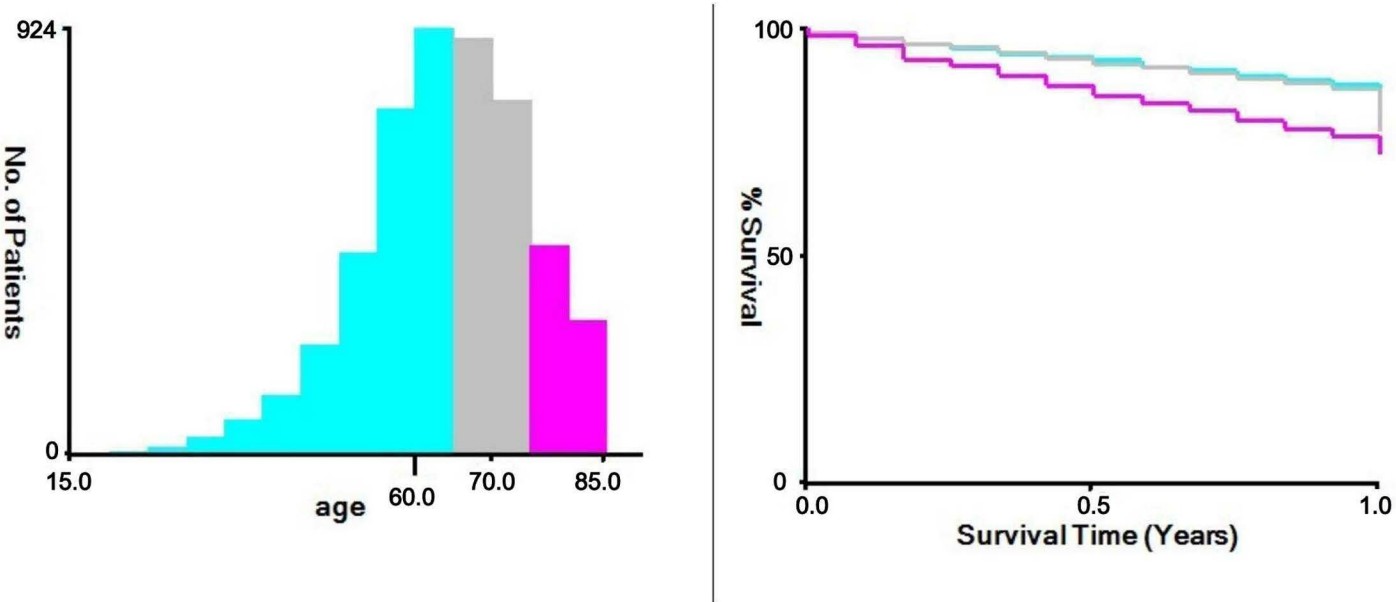

**Fig 2. The appropriate cutoff values of age was assessed by X-tile analysis: The appropriate cutoff values of age were 64 and 74 years.**

logistic regression model. Risk factors for stage III-IV endometrial cancer-specific early death were identified and used to construct the preliminary nomogram prediction model. The goodness-of-fit of the logistic regression model was examined using the Hosmer-Lemeshow test, by comparing observed and predicted outcomes in risk groups. The receiver operating characteristic (ROC) curve was used to assess the model's discrimination. The calibration curve was used to evaluate the model's calibration. The Decision Curve Analysis (DCA) curve evaluates the clinical utility of predictive models. The significance level was set at $\alpha=0.05$.

## Results

### General information of the study population

Based on the inclusion and exclusion criteria, a total of 5,148 patients with stage III-IV endometrial cancer were selected for the study. The dataset was randomly divided into a training cohort(n = 3603;70%) and a validation cohort(n = 1545;30%). Among 5,148 patients, there were 1294 early deaths, resulting in an early mortality rate of 25.1%. Of these, 352 were cancer-related, with a cancer-specific early mortality rate of 6.84%, accounting for 27.2% of all early deaths. Chi-square tests for R × C contingency tables revealed no statistically significant differences in clinicopathologic characteristics between the training and validation cohorts ($P > 0.05$), as detailed in (Table 1).

### Identification of risk factors for cancer-specific early death

The results of logistic regression analysis of the training cohort showed that tumor diameter, histological classification, histology grade, T-stage, M-stage, surgery, radiotherapy, chemotherapy, brain metastasis, and liver metastasis were the influencing factors for the occurrence of cancer-specific early death in endometrial cancer patients with stage III-IV endometrial cancer ($P < 0.05$), risk factors for cancer-specific early death include: tumor size ≥101 cm (OR = 2.11, $P < 0.001$), other histological classifications (OR = 3.11, $P < 0.001$), high histological tumor grades (G3) (OR = 2.68, $P = 0.007$), T3-T4 stage (OR = 1.84,

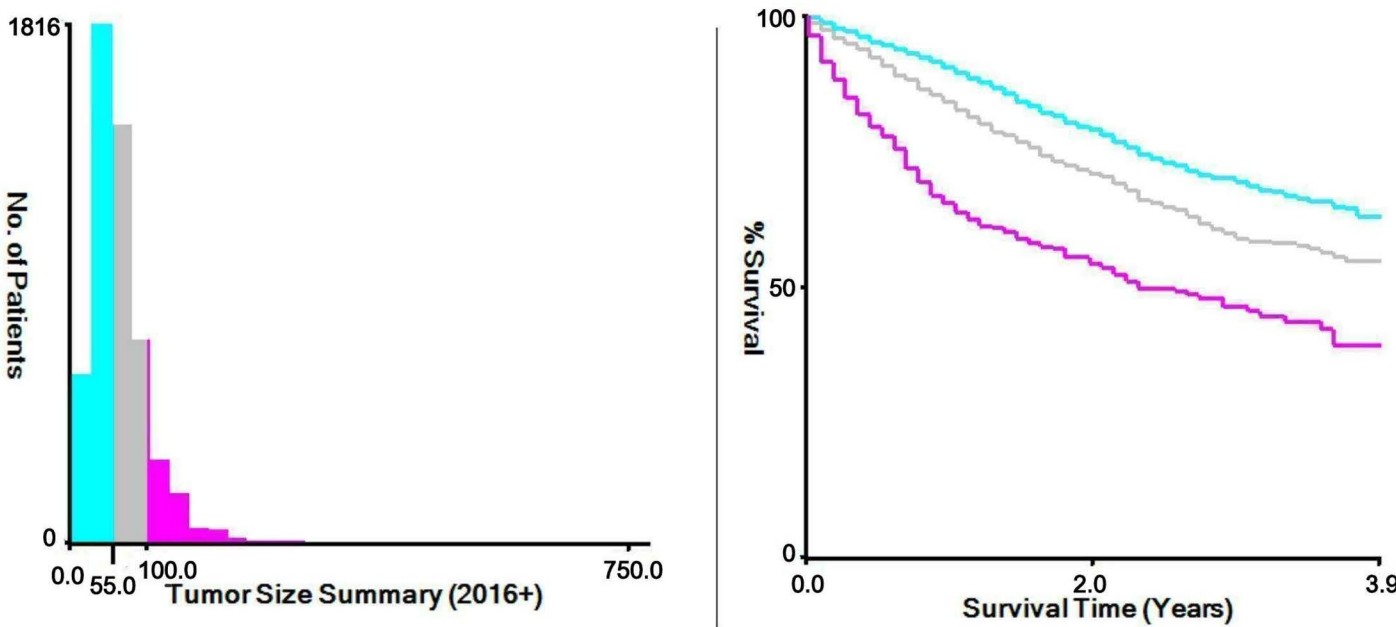

**Fig 3. The appropriate cutoff values of tumor size was assessed by X-tile analysis: The appropriate cutoff values of tumor size were 55 and 100 mm.**

$P = 0.004$), M1 stage (OR = 2.05, $P < 0.001$), liver metastasis (OR = 2.21, $P = 0.005$), and brain metastasis (OR = 8.08, $P < 0.001$). Conversely, protective factors include: surgery (hysterectomy (OR = 0.13, $P = 0.012$), radical surgery (OR = 0.21, $P < 0.001$)), radiation therapy (OR = 0.40, $P < 0.001$), and chemotherapy (OR = 0.31, $P < 0.001$), as shown in (Table 2). The results the of multifactor model covariance diagnosis showed that the variance inflation factor values of each variable were less than 10 (S1 Table), indicating that there was no multicollinearity among the independent variables of the model, suggesting that the results were stable and reliable. The results of the goodness-of-fit test of the cancer-specific early death model showed that $\chi 2 = 4.31$, $P = 0.828$, indicating that the models had high goodness-of-fit.

## Preliminary Model Development and Validation

According to the results of multifactorial logistic analysis, the factors with $P < 0.05$ were used to construct a preliminary nomogram model for predicting advanced endometrial cancer-specific early death. A vertical line is drawn from each prediction variable to the "Score" axis. Each prediction variable is then assigned the corresponding points shown by the intersection of the vertical line with the "Score" axis. The nomogram graphs summed up the scores of each predictor variable of the patients to calculate the total points, which is used to draw another vertical line from the "Total Points" axis to the probability axes, and the corresponding values underneath could be obtained to get the probability of the patient's cancer-specific early death, which can be seen in (Fig 4).

The ROC curve and its area under curve (AUC) were used to evaluate the differentiation of the preliminary model, and the AUC value of the cancer-specific early death prediction model in the training cohort was 0.89 [95% CI (0.87, 0.91)], while that of the validation cohort cancer-specific early death prediction model was 0.88 [95% CI (0.84, 0.91)]as shown in (Fig 5). Validation of the nomogram using a scatter plot of the actual probability

**Table 1. Comparison of clinicopathological characteristics of advanced EC patients between training and validation cohorts [n (%)].**

| Variables | Total (n = 5148) | test (n = 1545) | train (n = 3603) | Statistic | P |
|---|---|---|---|---|---|
| Age, n(%) | | | | $\chi^2 = 1.59$ | 0.451 |
| ≤64 | 2585 (50.21) | 781 (50.55) | 1804 (50.07) | | |
| 65–74 | 1669 (32.42) | 484 (31.33) | 1185 (32.89) | | |
| ≥75 | 894 (17.37) | 280 (18.12) | 614 (17.04) | | |
| Race, n(%) | | | | $\chi^2 = 0.98$ | 0.806 |
| White | 3832 (74.44) | 1146 (74.17) | 2686 (74.55) | | |
| Black | 649 (12.61) | 204 (13.20) | 445 (12.35) | | |
| Asian or Pacific Islander | 615 (11.95) | 181 (11.72) | 434 (12.05) | | |
| American Indian/Alaska Native | 52 (1.01) | 14 (0.91) | 38 (1.05) | | |
| Marital status, n(%) | | | | $\chi^2 = 0.05$ | 0.831 |
| Married | 2464 (47.86) | 743 (48.09) | 1721 (47.77) | | |
| single | 2684 (52.14) | 802 (51.91) | 1882 (52.23) | | |
| Tumor Size, n(%) | | | | $\chi^2 = 0.15$ | 0.927 |
| ≤55 | 2700 (52.45) | 804 (52.04) | 1896 (52.62) | | |
| 56–100 | 1835 (35.64) | 556 (35.99) | 1279 (35.50) | | |
| ≥101 | 613 (11.91) | 185 (11.97) | 428 (11.88) | | |
| Classification, n(%) | | | | $\chi^2 = 0.70$ | 0.703 |
| Endometrioid carcinoma | 3610 (70.12) | 1082 (70.03) | 2528 (70.16) | | |
| Serous cystadenocarcinoma | 1342 (26.07) | 399 (25.83) | 943 (26.17) | | |
| others | 196 (3.81) | 64 (4.14) | 132 (3.66) | | |
| Grade, n(%) | | | | $\chi^2 = 1.42$ | 0.702 |
| G1 | 902 (17.52) | 269 (17.41) | 633 (17.57) | | |
| G2 | 1239 (24.07) | 374 (24.21) | 865 (24.01) | | |
| G3 | 1772 (34.42) | 517 (33.46) | 1255 (34.83) | | |
| Grade Unknown | 1235 (23.99) | 385 (24.92) | 850 (23.59) | | |
| T, n(%) | | | | $\chi^2 = 0.05$ | 0.974 |
| T0–T2 | 1837 (35.68) | 553 (35.79) | 1284 (35.64) | | |
| T3–T4 | 3201 (62.18) | 960 (62.14) | 2241 (62.20) | | |
| Tx | 110 (2.14) | 32 (2.07) | 78 (2.16) | | |
| N, n(%) | | | | $\chi^2 = 3.06$ | 0.382 |
| N0 | 2004 (38.93) | 608 (39.35) | 1396 (38.75) | | |
| N1 | 2190 (42.54) | 638 (41.29) | 1552 (43.08) | | |
| N2 | 744 (14.45) | 240 (15.53) | 504 (13.99) | | |
| Nx | 210 (4.08) | 59 (3.82) | 151 (4.19) | | |
| M, n(%) | | | | $\chi^2 = 0.47$ | 0.491 |
| M0 | 3676 (71.41) | 1093 (70.74) | 2583 (71.69) | | |
| M1 | 1472 (28.59) | 452 (29.26) | 1020 (28.31) | | |
| Surgery, n(%) | | | | $\chi^2 = 2.26$ | 0.324 |
| no/local excision | 465 (9.03) | 153 (9.90) | 312 (8.66) | | |
| hysterectomy | 81 (1.57) | 26 (1.68) | 55 (1.53) | | |
| radical surgery | 4602 (89.39) | 1366 (88.41) | 3236 (89.81) | | |
| Lymph node, n(%) | | | | $\chi^2 = 2.63$ | 0.105 |
| no | 952 (18.49) | 265 (17.15) | 687 (19.07) | | |
| yes | 4196 (81.51) | 1280 (82.85) | 2916 (80.93) | | |
| Radiation, n(%) | | | | $\chi^2 = 0.32$ | 0.571 |
| no | 2610 (50.70) | 774 (50.10) | 1836 (50.96) | | |
| yes | 2538 (49.30) | 771 (49.90) | 1767 (49.04) | | |

*(Continued)*

**Table 1.** (Continued)

| Variables | Total (n = 5148) | test (n = 1545) | train (n = 3603) | Statistic | P |
|---|---|---|---|---|---|
| Chemotherapy, n(%) | | | | $\chi^2 = 0.16$ | 0.691 |
| no | 1158 (22.49) | 353 (22.85) | 805 (22.34) | | |
| yes | 3990 (77.51) | 1192 (77.15) | 2798 (77.66) | | |
| Systemic, n(%) | | | | $\chi^2 = 0.28$ | 0.596 |
| no | 1334 (25.91) | 408 (26.41) | 926 (25.70) | | |
| yes | 3814 (74.09) | 1137 (73.59) | 2677 (74.30) | | |
| Bone, n(%) | | | | $\chi^2 = 0.25$ | 0.617 |
| No/Unknown | 5007 (97.26) | 1500 (97.09) | 3507 (97.34) | | |
| Yes | 141 (2.74) | 45 (2.91) | 96 (2.66) | | |
| Brain, n(%) | | | | $\chi^2 = 0.09$ | 0.765 |
| No/Unknown | 5114 (99.34) | 1534 (99.29) | 3580 (99.36) | | |
| Yes | 34 (0.66) | 11 (0.71) | 23 (0.64) | | |
| Liver, n(%) | | | | $\chi^2 = 1.27$ | 0.261 |
| No/Unknown | 5008 (97.28) | 1509 (97.67) | 3499 (97.11) | | |
| Yes | 140 (2.72) | 36 (2.33) | 104 (2.89) | | |
| Lung, n(%) | | | | $\chi^2 = 0.96$ | 0.328 |
| No/Unknown | 4784 (92.93) | 1444 (93.46) | 3340 (92.70) | | |
| Yes | 364 (7.07) | 101 (6.54) | 263 (7.30) | | |
| All Death, n(%) | | | | $\chi^2 = 0.00$ | 0.980 |
| Alive | 3854 (74.86) | 1157 (74.89) | 2697 (74.85) | | |
| Dead | 1294 (25.14) | 388 (25.11) | 906 (25.15) | | |
| Cancer Death, n(%) | | | | $\chi^2 = 1.01$ | 0.314 |
| Alive | 4796 (93.16) | 1431 (92.62) | 3365 (93.39) | | |
| Dead | 352 (6.84) | 114 (7.38) | 238 (6.61) | | |

$\chi^2$: Chi-square test.

(Y-axis) against the predicted probability (X-axis) showed that the calibrations are close to the 45°line, which indicates that the predictions of the model were well in line with the actual results. This suggests that the model has good clinical utility, as shown in (Fig 6). The Decision Curve Analysis (DCA) curve ranges from 0.02 to 0.78 in the training cohort and from 0.02 to 0.73 in the validation cohort. Within these ranges, the DCA curve lies above the None and All null lines, indicating moderate effectiveness of the model. Outside these ranges, specifically below 0.02 and above 0.78 for the training cohort and above 0.73 for the validation cohort, the DCA curve approaches the None and All null lines, suggesting reduced effectiveness (Fig 7).

## Discussion

With changes in lifestyle and dietary habits, the incidence of endometrial cancer is rising, and the age at diagnosis is trending younger. Despite significant advancements in surgical techniques and systemic drug therapies for advanced endometrial cancer, managing the disease remains challenging. Most research has focused on clinical treatment, with less emphasis on early mortality. This study contributes to identifying key factors associated with early death in advanced endometrial cancer, providing insights that could help improve patient management strategies and reduce early mortality.

**Table 2. The univariate and multivariate logistic regression analysis of cancer-specific early death.**

| Variables | Univariate analysis | | | | | Multivariate analysis | | | | |
|---|---|---|---|---|---|---|---|---|---|---|
| | β | S.E | Z | *P* | OR (95%CI) | β | S.E | Z | *P* | OR (95%CI) |
| Age | | | | | | | | | | |
| ≤64 | | | | | 1.00 (Reference) | | | | | 1.00 (Reference) |
| 65–74 | 0.17 | 0.16 | 1.08 | 0.280 | 1.18 (0.87–1.61) | 0.05 | 0.19 | 0.24 | 0.808 | 1.05 (0.72–1.51) |
| ≥75 | 0.63 | 0.17 | 3.72 | **<.001** | 1.88 (1.35–2.62) | 0.23 | 0.21 | 1.08 | 0.281 | 1.26 (0.83–1.92) |
| Race | | | | | | | | | | |
| White | | | | | 1.00 (Reference) | | | | | 1.00 (Reference) |
| Black | 0.47 | 0.18 | 2.60 | **0.009** | 1.59 (1.12–2.26) | 0.26 | 0.22 | 1.21 | 0.228 | 1.30 (0.85–1.98) |
| Asian or Pacific Islander | −0.09 | 0.22 | −0.43 | 0.671 | 0.91 (0.59–1.40) | −0.06 | 0.25 | −0.24 | 0.808 | 0.94 (0.58–1.54) |
| American Indian/Alaska Native | −0.91 | 1.02 | −0.90 | 0.371 | 0.40 (0.05–2.95) | −1.04 | 1.09 | −0.95 | 0.340 | 0.35 (0.04–2.99) |
| Marital status | | | | | | | | | | |
| Married | | | | | 1.00 (Reference) | | | | | |
| single | 0.07 | 0.13 | 0.49 | 0.621 | 1.07 (0.82–1.39) | | | | | |
| Tumor Size | | | | | | | | | | |
| ≤55 | | | | | 1.00 (Reference) | | | | | 1.00 (Reference) |
| 56–100 | 0.65 | 0.16 | 4.02 | **<.001** | 1.91 (1.39–2.63) | 0.33 | 0.19 | 1.77 | 0.076 | 1.39 (0.97–2.00) |
| ≥101 | 1.65 | 0.17 | 9.45 | **<.001** | 5.22 (3.71–7.35) | 0.75 | 0.22 | 3.44 | **<.001** | 2.11 (1.38–3.22) |
| Classification | | | | | | | | | | |
| Endometrioid carcinoma | | | | | 1.00 (Reference) | | | | | 1.00 (Reference) |
| Serous cystadenocarcinoma | 0.26 | 0.15 | 1.71 | 0.087 | 1.30 (0.96–1.77) | −0.07 | 0.20 | −0.34 | 0.734 | 0.93 (0.63–1.38) |
| others | 1.79 | 0.22 | 8.25 | **<.001** | 6.01 (3.92–9.20) | 1.14 | 0.27 | 4.18 | **<.001** | 3.11 (1.83–5.30) |
| Grade | | | | | | | | | | |
| G1 | | | | | 1.00 (Reference) | | | | | 1.00 (Reference) |
| G2 | 0.02 | 0.42 | 0.06 | 0.953 | 1.02 (0.45–2.32) | −0.07 | 0.44 | −0.15 | 0.879 | 0.94 (0.40–2.20) |
| G3 | 1.51 | 0.34 | 4.47 | **<.001** | 4.53 (2.33–8.78) | 0.98 | 0.37 | 2.68 | **0.007** | 2.68 (1.30–5.51) |
| Grade Unknown | 2.41 | 0.33 | 7.25 | **<.001** | 11.15 (5.81–21.40) | 0.63 | 0.41 | 1.54 | 0.124 | 1.87 (0.84–4.16) |
| T | | | | | | | | | | |
| T0–T2 | | | | | 1.00 (Reference) | | | | | 1.00 (Reference) |
| T3–T4 | 0.89 | 0.18 | 5.08 | **<.001** | 2.44 (1.73–3.45) | 0.61 | 0.21 | 2.89 | **0.004** | 1.84 (1.22–2.79) |
| Tx | 2.64 | 0.29 | 9.12 | **<.001** | 13.95 (7.92–24.57) | −0.01 | 0.37 | −0.02 | 0.984 | 0.99 (0.49–2.03) |
| N | | | | | | | | | | |
| N0 | | | | | 1.00 (Reference) | | | | | 1.00 (Reference) |
| N1 | −0.44 | 0.17 | −2.63 | **0.008** | 0.65 (0.47–0.89) | −0.05 | 0.20 | −0.24 | 0.812 | 0.95 (0.65–1.41) |
| N2 | 0.32 | 0.19 | 1.65 | 0.099 | 1.37 (0.94–2.00) | 0.44 | 0.23 | 1.94 | 0.052 | 1.55 (1.00–2.42) |
| Nx | 1.50 | 0.22 | 6.84 | **<.001** | 4.49 (2.92–6.90) | 0.51 | 0.27 | 1.91 | 0.056 | 1.66 (0.99–2.81) |
| M | | | | | | | | | | |
| M0 | | | | | 1.00 (Reference) | | | | | 1.00 (Reference) |
| M1 | 1.75 | 0.14 | 12.23 | **<.001** | 5.73 (4.33–7.59) | 0.72 | 0.20 | 3.52 | **<.001** | 2.05 (1.38–3.06) |
| Surgery | | | | | | | | | | |
| no/local excision | | | | | 1.00 (Reference) | | | | | 1.00 (Reference) |
| hysterectomy | −2.64 | 0.73 | −3.62 | **<.001** | 0.07 (0.02–0.30) | −2.01 | 0.81 | −2.50 | **0.012** | 0.13 (0.03–0.65) |
| radical surgery | −2.55 | 0.15 | −17.10 | **<.001** | 0.08 (0.06–0.10) | −1.57 | 0.29 | −5.33 | **<.001** | 0.21 (0.12–0.37) |
| Lymph node | | | | | | | | | | |
| no | | | | | 1.00 (Reference) | | | | | 1.00 (Reference) |
| yes | 1.31 | 0.27 | 4.85 | **<.001** | 3.71 (2.18–6.30) | −0.14 | 0.31 | −0.46 | 0.646 | 0.87 (0.48–1.58) |
| Radiation | | | | | | | | | | |
| no | | | | | 1.00 (Reference) | | | | | 1.00 (Reference) |

*(Continued)*

**Table 2.** (Continued)

| Variables | Univariate analysis | | | | | Multivariate analysis | | | | |
|---|---|---|---|---|---|---|---|---|---|---|
| | β | S.E | Z | *P* | OR (95%CI) | β | S.E | Z | *P* | OR (95%CI) |
| yes | −1.39 | 0.16 | −8.49 | **<.001** | 0.25 (0.18–0.34) | −0.91 | 0.20 | −4.47 | **<.001** | 0.40 (0.27–0.60) |
| Chemotherapy | | | | | | | | | | |
| no | | | | | 1.00 (Reference) | | | | | 1.00 (Reference) |
| yes | −1.68 | 0.14 | −12.19 | **<.001** | 0.19 (0.14–0.24) | −1.17 | 0.26 | −4.58 | **<.001** | 0.31 (0.19–0.51) |
| Systemic | | | | | | | | | | |
| no | | | | | 1.00 (Reference) | | | | | 1.00 (Reference) |
| yes | −2.22 | 0.15 | −14.70 | **<.001** | 0.11 (0.08–0.15) | −0.56 | 0.29 | −1.89 | 0.059 | 0.57 (0.32–1.02) |
| Bone | | | | | | | | | | |
| No/Unknown | | | | | 1.00 (Reference) | | | | | 1.00 (Reference) |
| Yes | 1.51 | 0.25 | 5.98 | **<.001** | 4.53 (2.76–7.43) | 0.35 | 0.32 | 1.10 | 0.270 | 1.42 (0.76–2.63) |
| Brain | | | | | | | | | | |
| No/Unknown | | | | | 1.00 (Reference) | | | | | 1.00 (Reference) |
| Yes | 2.43 | 0.43 | 5.69 | **<.001** | 11.31 (4.91–26.07) | 2.09 | 0.51 | 4.10 | **<.001** | 8.08 (2.98–21.93) |
| Liver | | | | | | | | | | |
| No/Unknown | | | | | 1.00 (Reference) | | | | | 1.00 (Reference) |
| Yes | 2.06 | 0.22 | 9.31 | **<.001** | 7.85 (5.09–12.10) | 0.79 | 0.28 | 2.82 | **0.005** | 2.21 (1.27–3.82) |
| Lung | | | | | | | | | | |
| No/Unknown | | | | | 1.00 (Reference) | | | | | 1.00 (Reference) |
| Yes | 1.71 | 0.16 | 10.41 | **<.001** | 5.55 (4.02–7.66) | −0.04 | 0.23 | −0.18 | 0.855 | 0.96 (0.61–1.51) |

OR: Odds Ratio, CI: Confidence Interval

## Risk factors for cancer-specific early death in advanced endometrial cancer

This study analyzed the risk factors for cancer-specific early death and constructed, validated, and evaluated a Nomogram prediction model based on the inclusion and exclusion criteria of 5148 patients with advanced endometrial cancer extracted from the SEER database. The results of this study showed that tumor diameter, histological classification, histology grade, T-stage, M-stage, surgery, radiotherapy, chemotherapy, concomitant brain metastasis and liver metastasis were the influencing factors for cancer-specific early death in patients with advanced endometrial cancer.

The results of this study indicate that endometrial cancer patients with larger tumor diameters (≥101 mm) have a significantly worse prognosis. This is likely due to the fact that larger tumors are often associated with higher tumor grades, deeper myometrial invasion, and elevated rates of metastasis. An increased tumor diameter may reflect more rapid tumor cell proliferation and a consequently higher risk of lymph node metastasis [6,7]. Additionally, our study found that the overall survival rate for other histologic subtypes, including clear cell carcinoma, was significantly lower compared to that of endometrioid adenocarcinoma, aligning with the findings of Liu et al. [8]. The high histological tumor grades(G3) as an independent risk factor for early death in advanced endometrial cancer supports previous research indicating that lower degrees of tissue differentiation are associated with more malignant and invasive tumors, increasing the risk of both local and distant metastases and adversely affecting patient survival [9]. Moreover, higher staging was strongly correlated with poorer prognosis, consistent with the prognostic value of the TNM staging system.

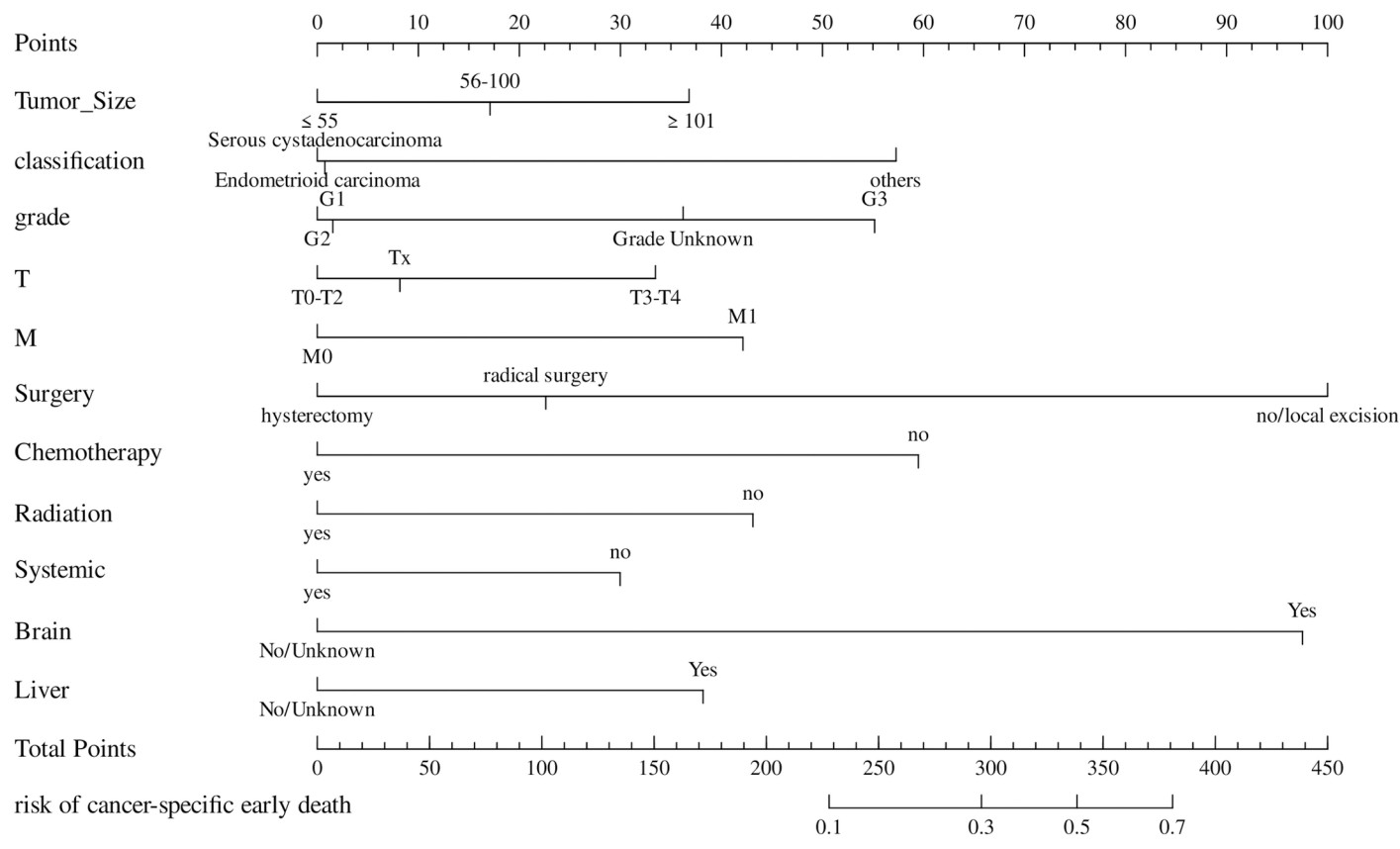

**Fig 4. The preliminary nomograms of the cancer-specific early death in patients with stage III-IV endometrial carcinoma.**

Specifically, higher T-stage, N-stage, and M-stage were all associated with worse outcomes for patients.

There is ongoing academic debate regarding the necessity of primary site surgical treatment for patients with advanced endometrial cancer. The findings of this study suggest that among the available surgical options, hysterectomy and radical surgery significantly reduce the risk of cancer-specific early mortality, with radical surgery demonstrating a particularly significant survival advantage. These results imply that hysterectomy or radical surgery is an effective strategy for prolonging survival in patients who are medically fit and meet surgical criteria. A retrospective analysis by Yutaka et al. [10] involving 33 patients with stage IV endometrial cancer found that those undergoing tumor debulking surgery (residual lesions ≤ 2 cm) had markedly better median progression-free survival (PFS) and overall survival (OS) compared to patients with residual lesions > 2 cm. This improvement was consistent across both intra-abdominal and extra-abdominal metastatic cases (P < 0.05). Additionally, another observational study of 102 patients with advanced endometrial cancer recommended that debulking surgery should aim for no residual tumor (R0), regardless of histologic subtype [11].For patients with advanced (stage III to IV) endometrial cancer, a comprehensive evaluation by a multidisciplinary team may indicate that surgical intervention, including complete lesion excision and removal of enlarged lymph nodes, could enhance survival and quality of life if feasible. In stage IV cases, palliative total hysterectomy with bilateral adnexectomy may also be considered when appropriate, based on multidisciplinary consultation [12,13].

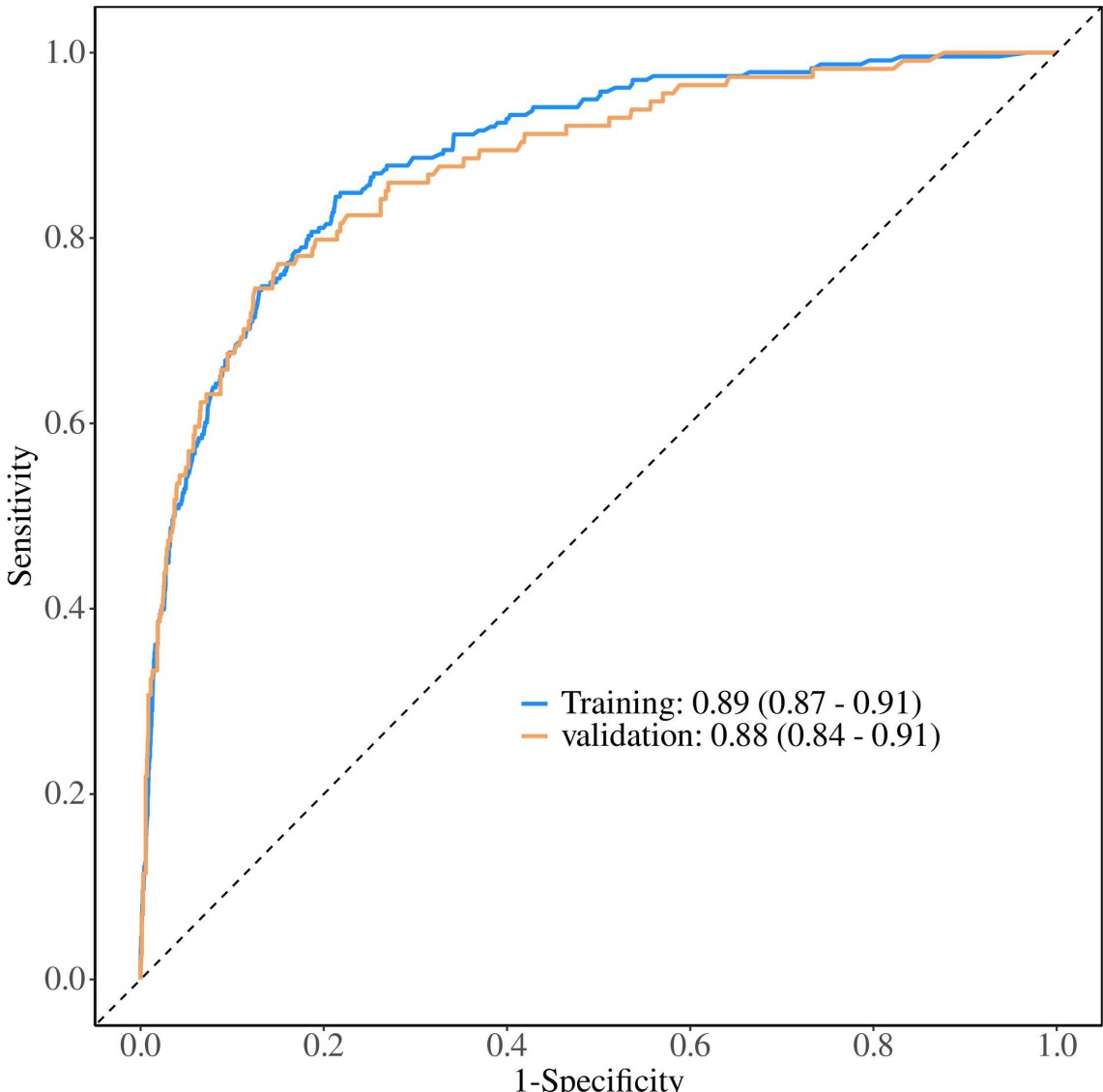

**Fig 5. The receiver operating characteristic (ROC) curve for nomogram.** The **x-axis** represents the false positive rate (FPR), and the **y-axis** represents the true positive rate (TPR).

The results of this study also indicate that the administration of chemotherapy and radiotherapy serves as a protective factor against early death in patients with advanced endometrial cancer. Research by Eskinder et al. [14] has suggested that postoperative adjuvant radiotherapy or chemotherapy can effectively manage local tumor recurrence, thereby enhancing long-term survival rates. This reinforces the critical role of chemoradiotherapy as a key component in the management of advanced endometrial cancer. Additionally, this study identified that patients with liver and brain metastases have a higher likelihood of experiencing cancer-specific early death. This suggests that liver and brain metastases are associated with an increased risk of early mortality in metastatic endometrial cancer. These findings are consistent with those of Michae et al. [15], who reported that distant metastases, particularly liver metastases, are a major cause of death in endometrial cancer patients. These results highlight

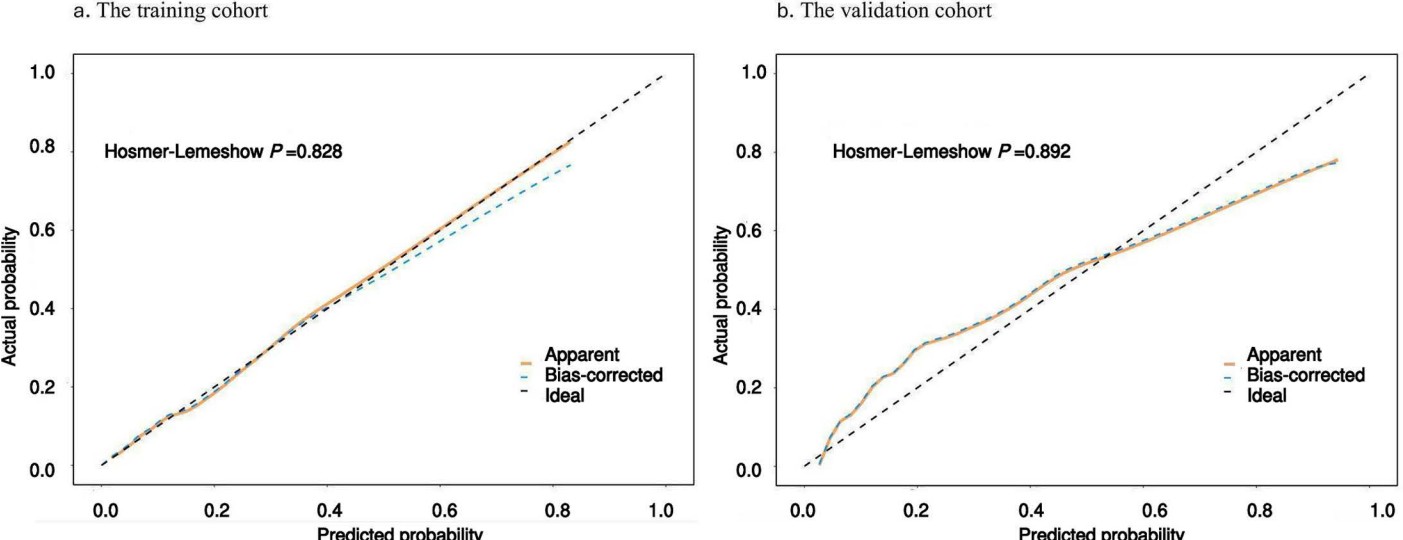

**Fig 6. Calibration curve for cancer-specific early death: (a) The training cohort, (b) The validation cohort.** The **x-axis** represents the predicted probabilities, while the **y-axis** shows the actual observed probabilities. The **apparent line** shows the model's calibration on the training data, the **bias-corrected line** adjusts for overfitting to reflect performance on new data, and the **ideal line** (45-degree diagonal) represents perfect calibration. The Hosmer-Lemeshow (HL) test evaluates calibration by comparing observed and predicted outcomes in risk groups, with a higher p-value indicating better calibration.

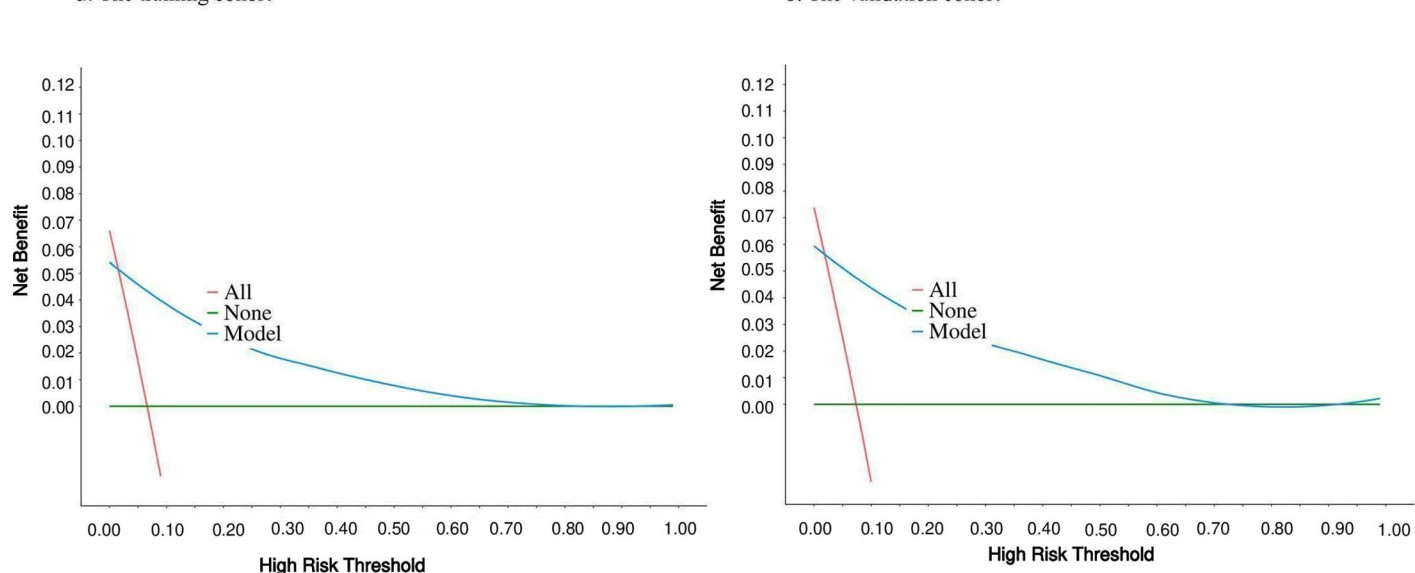

**Fig 7. The decision curve analysis (DCA) curve for nomogram: (a) The training cohort, (b) The validation cohort.** The x-axis represents the threshold probability at which a decision is made to intervene or not, while the y-axis represents the net benefit, showing the overall benefit of using the model at different threshold probabilities. the **model line** represents the ideal model's perfect prediction. The **None line** shows the net benefit of not treating any patients, while the **All line** shows the net benefit of treating all patients.

the necessity for more personalized and aggressive treatment strategies for patients with advanced endometrial cancer, particularly those with liver and brain metastases, to mitigate the risk of early death and improve overall survival.

Additionally, this study suggests that lymph node dissection does not significantly affect early survival in patients with advanced endometrial cancer. This observation aligns with findings from Benedetti et al. [16], who also reported that lymph node dissection was not associated with improved survival outcomes in endometrial cancer patients. The clinical utility of lymph node dissection may warrant reevaluation in the context of treatment planning for advanced endometrial cancer. While lymph node dissection can be informative for assessing the extent of disease dissemination, its direct impact on patient survival may not be as substantial as previously anticipated. Consequently, further research is needed to elucidate the specific role and optimal indications for lymph node dissection in the management of endometrial cancer.

## Treatment of advanced endometrial cancer

Aggressive, comprehensive treatment is widely regarded as an effective strategy to mitigate the risk of early death in patients with advanced endometrial cancer. Currently, there is no standardized protocol for treating advanced endometrial cancer, which may include chemotherapy, radiotherapy, targeted therapy, immunotherapy, and surgery. A multidisciplinary team (MDT) is essential for formulating a personalized treatment plan, as combination therapy represents the primary strategy for managing advanced cases. This approach integrates both localized treatments (e.g., surgery, radiotherapy, and other interventions) and systemic treatments (e.g., chemotherapy, endocrine therapy, immunotherapy, and targeted therapies). However, in our study indicated that systemic therapy did not significantly influence early survival outcomes in patients with advanced endometrial cancer. This finding may be attributed to several factors, including limited adoption of new therapies during the study period and insufficient clinical experience. Additionally, patient selection and treatment adherence issues could have impacted early survival outcomes. Recent advances in the clinical management of advanced or recurrent endometrial cancer have been driven by intensified research into tumor molecular biology and the introduction of targeted agents and immune checkpoint inhibitors (ICIs). A retrospective study demonstrated that the combination of paclitaxel, carboplatin, and bevacizumab achieved significant efficacy in patients with recurrent or inoperable stage IVB endometrial cancer, with a median progression-free survival (mPFS) of 20 months, median overall survival (mOS) of 56 months, and an objective response rate (ORR) of 82.8% for first-line patients. The ORR for second-line patients was reported at 87.5%(14).With the advent of new targeted drugs and the maturation of immunotherapy, combination regimens incorporating immunotherapy and chemotherapy are anticipated to become the standard first-line treatment for advanced endometrial cancer. Notably, the NRG-GY108 and RUBY studies published in the New England Journal of Medicine in 2023 demonstrated positive results in phase III randomized controlled trials of pembrolizumab and doxorubicin, respectively, for stage III, IV, or recurrent endometrial cancer [17]. These studies found that progression-free survival (PFS) was superior in the immunotherapy groups compared to those receiving chemotherapy alone, across both mismatch repair-deficient (dMMR) and mismatch repair-proficient (pMMR) populations.

In this study, our primary goal was to thoroughly explore and identify the key risk factors that influence early mortality in advanced endometrial cancer. Through meticulous data analysis, we successfully identified a series of significant risk factors and constructed a preliminary predictive model to analyze these factors. However, this article has limitations. Firstly, the SEER database does not include general health status information of patients,

such as body mass index (BMI), family history of malignancies, or underlying conditions such as diabetes mellitus, hypertension, and cardiac disorders. The absence of these factors restricts our ability to fully evaluate their impact on patient prognosis. Secondly, the study lacks detailed records regarding the specific regimens and durations of radiotherapy and chemotherapy, as well as the specifics of targeted or immunotherapy. Such treatment-related data are crucial for accurately assessing treatment efficacy and prognosis. To address these limitations, future research should involve multicenter prospective clinical cohort studies designed to gather more comprehensive patient information, including previously unre-corded health indicators. Enhancing data completeness will improve the stability and pre-dictive accuracy of study models. Given the current dearth of prediction models for patients with advanced endometrial cancer, our study may serve as a pioneering effort. We encourage future researchers to continue exploring and validating our model and to build upon it for further development.

In summary, this study utilized the SEER database to gather demographic, clinicopatho-logic, and follow-up data for patients with advanced endometrial cancer. The analysis iden-tified risk factors associated with cancer-specific early deaths and developed a corresponding nomogram prediction model. These findings assist clinicians in early identification of high-risk patients and offer a scientific foundation for creating individualized treatment strategies.

## Supporting information

**S1 Table. The variance inflation factor values of each variable.**
(DOCX)

## Author contributions

**Data curation:** Qiao Liu, Yi Tang.

**Formal analysis:** Yi Tang, Chuqiang Shu.

**Funding acquisition:** Yi Tang, Dan Jiang.

**Investigation:** Qiao Liu.

**Methodology:** Dan Jiang.

**Visualization:** Qi Tian, Guang Li.

**Writing – original draft:** Jing Yang.

**Writing – review & editing:** Jing Yang, Qi Tian, Chuqiang Shu.

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
