## [Decision Letter · Decision Letter 0]

15 Oct 2024

PONE-D-24-32320The nomograms for predicting cancer-specific early death in patients with advanced endometrial cancer: A population-based studyPLOS ONE

Dear Dr. Shu,

Thank you for submitting your manuscript to PLOS ONE. After careful consideration, we feel that it has merit but does not fully meet PLOS ONE’s publication criteria as it currently stands. Therefore, we invite you to submit a revised version of the manuscript that addresses the points raised during the review process.

**ACADEMIC EDITOR: **

Please make sure that

1. The study presents the results of original research.

2. Results reported have not been published elsewhere.

3. Experiments, statistics, and other analyses are performed to a high technical standard and are described in sufficient detail.

4. Conclusions are presented in an appropriate fashion and are supported by the data.

5. The article is presented in an intelligible fashion and is written in standard English.

6. The research meets all applicable standards for the ethics of experimentation and research integrity.

7. The article adheres to appropriate reporting guidelines and community standards for data availability.

8. The manuscript conforms to STROBE reporting guidelines.

We look forward to receiving your revised manuscript.

Kind regards,

Bella Stevanny

Academic Editor

PLOS ONE

5. We notice that your supplementary table is included in the manuscript file. Please remove them and upload them with the file type 'Supporting Information'. Please ensure that each Supporting Information file has a legend listed in the manuscript after the references list.

Reviewers' comments:

Reviewer's Responses to Questions

**Comments to the Author**

1. Is the manuscript technically sound, and do the data support the conclusions?

Reviewer #1: Yes

Reviewer #2: Yes

Reviewer #3: Yes

2. Has the statistical analysis been performed appropriately and rigorously? 

Reviewer #1: Yes

Reviewer #2: Yes

Reviewer #3: Yes

3. Have the authors made all data underlying the findings in their manuscript fully available?

Reviewer #1: Yes

Reviewer #2: Yes

Reviewer #3: Yes

4. Is the manuscript presented in an intelligible fashion and written in standard English?

Reviewer #1: Yes

Reviewer #2: Yes

Reviewer #3: Yes

5. Review Comments to the Author

Reviewer #1: The manuscript reports on an interesting study that collected clinical data from patients with advanced endometrial cancer in the SEER database from 2018 to 2021, randomly dividing all patients into a training cohort and a validation cohort in a ratio of 7:3. Multivariate logistic regression analysis was performed on the training cohort to screen out the risk factors of cancer-specific early death in patients with advanced endometrial cancer, and the nematogram model was further constructed for verification. However, there are still some problems in this study, the flow chart is rough and some arrows point wrong. In addition to the marital status of single and married patients, are there any other conditions, such as divorce and widowhood, etc., in the general information of patients? Are these people included in the study?

Reviewer #2: Thank you for introducing such a good and reliable method for predicting the mortalities associated with endometrial cancer which will be positively reflected on the councelling as well as managing those patients.

Reviewer #3: 1. This is a clear and concise article based on correct statistical methods on predicting cancer-specific early death in patients with advanced endometrial cancer.

2. While details explanation on main statistical methods were presented, there was no mention regarding any other clinical prediction model in the literature for advanced endometrial cancer. If there are other models, this should be mentioned in the introduction and compared in the discussion section. If none, this should also be mentioned that there is no prediction model for this specific patient’s group. There should be some information regarding prediction model in the introduction section.

3. The methods section does not specify the coding of the outcome variable for logistic regression: Was it early death or cancer-specific early death?

4. Table 1 shows that there is no variation between training and validation data sets in terms of considered variable and hence model performance was good in validation. However, it does not validate this model in real life heterogenous data set, which is more common.

5. In line 210: The study found that the early all-cause mortality rate for stage III-IV endometrial cancer was 25.1%, while the early cancer-specific mortality rate was 6.8%. Therefore, a large percentage of early death of cancer patients are due to non-cancer related causes including co-morbidities, which were not included in the model.

6. Co-morbidity data were absent in the used data set, which is a big limitation of this article in terms of the usefulness of this predictive model. Previous studies showed that comorbidity indices like Age-adjusted Charlson Comorbidity Index is a strong predictor of overall survival of early-stage endometrial cancer patients. (Al Feghali KA, Robbins JR, Mahan M, Burmeister C, Khan NT, Rasool N, Munkarah A, Elshaikh MA. Predictive Capacity of 3 Comorbidity Indices in Estimating Survival Endpoints in Women With Early-Stage Endometrial Carcinoma. Int J Gynecol Cancer. 2016 Oct;26(8):1455-60. doi: 10.1097/IGC.0000000000000802. PMID: 27488218). This article identifies the significant risk factors for early-stage endometrial cancer but has limited use as a prediction model. Therefore, the focus of this article could be more on identifying risk factors over prediction unless comorbidity variables are included in the models.

7. P in P-value should be italic.

8. CI in Table 2 should have either – sign or comma but not Tilda sign (~).

9. In line 118: “An ROC curve closer to 1 reflects a model with high TPR and low FPR, making it a better-performing model”. Sentences like this may be deleted depending on the background of the readers of this article.

6. PLOS authors have the option to publish the peer review history of their article (what does this mean? ). If published, this will include your full peer review and any attached files.

**Do you want your identity to be public for this peer review?** For information about this choice, including consent withdrawal, please see our Privacy Policy .

Reviewer #1: No

Reviewer #2: **Yes: ** Mohsen M A Abdelhafez

Reviewer #3: **Yes: ** Dr Shah-Jalal Sarker

---

## [Author Response · Author response to Decision Letter 1]

7 Nov 2024

Reviewer #3: 1. This is a clear and concise article based on correct statistical methods on predicting cancer-specific early death in patients with advanced endometrial cancer.

Thank you for your kind words and positive evaluation of our article. We appreciate your recognition of the importance of clear and concise analytical methods in predicting cancer-specific early deaths in patients with advanced endometrial cancer. Your feedback validates the accuracy and reliability of our research, encouraging us to continue our efforts to meet the needs of the medical community and improve patient care.

2. While details explanation on main statistical methods were presented, there was no mention regarding any other clinical prediction model in the literature for advanced endometrial cancer. If there are other models, this should be mentioned in the introduction and compared in the discussion section. If none, this should also be mentioned that there is no prediction model for this specific patient’s group. There should be some information regarding prediction model in the introduction section.

We would like to express our sincere gratitude for your meticulous review and valuable feedback. Your observation regarding the absence of mention of other clinical prediction models in the literature for advanced endometrial cancer is well-taken and highlights an oversight in our manuscript. We acknowledge the importance of referencing existing prediction models in the literature and comparing them in the discussion section to provide readers with a comprehensive understanding of the context and significance of our study.

Upon re-examining the relevant literature, we regret to find that, to the best of our knowledge, there are currently no other established clinical prediction models specifically for this patient group with advanced endometrial cancer. This may be attributed to the complexity and heterogeneity of the disease, which poses challenges in developing a universally applicable prediction model. However, this realization also presents our study with a unique opportunity to develop the first prediction model tailored to this specific patient cohort, thereby addressing a gap in the existing literature.

In response to your concern, we have added the following statement to the introduction section of our paper: " It should be highlighted that the current literature does not offer comprehensive predictive models specifically designed for patients with advanced endometrial cancer."

Additionally, in the discussion section, we have included a reflection on this state of affairs: "Given the current dearth of prediction models for patients with advanced endometrial cancer, our study may serve as a pioneering effort. We encourage future researchers to continue exploring and validating our model and to build upon it for further development."

We believe these additions will enhance the completeness and transparency of our article. We appreciate your valuable comments and look forward to your further feedback.

3. The methods section does not specify the coding of the outcome variable for logistic regression: Was it early death or cancer-specific early death?

Thank you for your careful review and insightful comments on our manuscript. You are correct in noting that we did not explicitly define whether "early death" or "cancer-specific early death" was used as the outcome variable in our logistic regression analysis within the methods section. This is indeed a critical detail that requires clarification.

In our study, we utilized "cancer-specific early death" as the outcome variable for our logistic regression model. This was defined as death due to endometrial cancer within six months of the initial pathological diagnosis. To address your concern and to avoid any ambiguity, we will revise the methods section to include a clear definition of our outcome variable.

Here is how we propose to revise the methods section to include this definition:

" The primary outcome variable for the logistic regression analysis was defined as 'cancer-specific early death,' which refers to death attributable to endometrial cancer within six months following the initial pathological diagnosis. "

By specifying our definition of 'cancer-specific early death' in this manner, we aim to enhance the clarity and transparency of our methods and to ensure that the outcome variable is clearly understood by readers. We believe this revision will address your concern and strengthen our manuscript.

We appreciate your valuable feedback and are grateful for the opportunity to improve the quality of our submission.

4. Table 1 shows that there is no variation between training and validation data sets in terms of considered variable and hence model performance was good in validation. However, it does not validate this model in real life heterogenous data set, which is more common.

We greatly appreciate your insightful comments and acknowledge the critical importance of external validation in the context of our predictive model for advanced endometrial cancer. Your feedback has prompted us to re-evaluate our approach and focus on the aspects that are most relevant to our study's objectives.

In light of the limitations you've pointed out, particularly the absence of detailed comorbidity data in the SEER database, we have decided to pivot our manuscript to emphasize the identification and analysis of key risk factors associated with early cancer-specific mortality in patients with advanced endometrial cancer, rather than the model's predictive performance.

Here are the adjustments we have made to address your concerns:

1. Refocused Manuscript Scope: We have revised the manuscript to underscore the importance of risk factors such as tumor size, histological type, tumor grade, and clinical stage. While these factors do not encompass the full spectrum of comorbidities, they remain significant for clinical decision-making and treatment strategy formulation.

2. Acknowledgment of Preliminary Nature: We have explicitly stated in the manuscript that our findings are preliminary and that our model serves as a foundation for future studies. This acknowledgment is prominently featured in the discussion section, setting the stage for more comprehensive analyses that will include comorbidity data.

3. Future Research Directions: We have outlined potential directions for future research, suggesting the incorporation of comorbidity data to enhance the model's predictive power and generalizability. We propose that future studies may benefit from multicenter, prospective designs or collaborative analyses with other databases to capture a more holistic view of patient health.

By refocusing on the risk factors and the preliminary nature of our findings, we aim to provide actionable insights for the clinical management of advanced endometrial cancer while also setting the stage for more robust validation studies in the future. We believe this approach aligns with the current data available and addresses the limitations inherent in our study design.

We are grateful for your valuable feedback and are committed to incorporating these considerations into our final manuscript. We are confident that these revisions will enhance the relevance and impact of our research within the scientific community.

5. In line 210: The study found that the early all-cause mortality rate for stage III-IV endometrial cancer was 25.1%, while the early cancer-specific mortality rate was 6.8%. Therefore, a large percentage of early death of cancer patients are due to non-cancer related causes including co-morbidities, which were not included in the model.

Thank you for highlighting this important aspect of our study. We appreciate your observation regarding the distinction between all-cause mortality and cancer-specific mortality in patients with stage III-IV endometrial cancer.

Specifically, The cancer-specific mortality rate for endometrial cancer is 6.8%，while the 25.1% all-cause mortality includes deaths related to other cancers and co-morbid factors.You are correct that a significant proportion of early deaths may be attributed to non-cancer related causes, including co-morbidities, which are not accounted for in our current model. This is indeed a limitation of our study, primarily due to the lack of detailed comorbidity data in the SEER database.

To address this limitation, we have taken the following steps:

Emphasis on Risk Factors: We have adjusted the manuscript to emphasize the identification of risk factors associated with cancer-specific early death. While our model does not include co-morbidities, we have focused on the risk factors that are available and can be used to inform clinical decision-making.

Acknowledgment of Limitations: We have explicitly acknowledged this limitation in the discussion section of our manuscript. We have clarified that the model's predictive ability may be limited in real-world settings where co-morbidities play a significant role in patient outcomes.

Future Research Directions: We have proposed that future research should aim to incorporate comorbidity data to enhance the model's predictive accuracy and applicability. This could involve multicenter, prospective studies or collaborations with other databases that contain more comprehensive patient health information.

Model Refinement: We are committed to refining our model in future studies by including additional variables and adjusting parameters to better capture the impact of co-morbidities on early mortality rates.

We appreciate your feedback and will ensure that these points are clearly articulated in the final version of our manuscript. We believe that by addressing this limitation and focusing on the risk factors we can control and measure, our research can still provide valuable insights into the early mortality rates of patients with advanced endometrial cancer.

6. Co-morbidity data were absent in the used data set, which is a big limitation of this article in terms of the usefulness of this predictive model. Previous studies showed that comorbidity indices like Age-adjusted Charlson Comorbidity Index is a strong predictor of overall survival of early-stage endometrial cancer patients. (Al Feghali KA, Robbins JR, Mahan M, Burmeister C, Khan NT, Rasool N, Munkarah A, Elshaikh MA. Predictive Capacity of 3 Comorbidity Indices in Estimating Survival Endpoints in Women With Early-Stage Endometrial Carcinoma. Int J Gynecol Cancer. 2016 Oct;26(8):1455-60. doi: 10.1097/IGC.0000000000000802. PMID: 27488218). This article identifies the significant risk factors for early-stage endometrial cancer but has limited use as a prediction model. Therefore, the focus of this article could be more on identifying risk factors over prediction unless comorbidity variables are included in the models.

We sincerely appreciate the keen observation you've made regarding the absence of comorbidity data in our study, a point that resonates with the heart of our research's limitations. Your reference to the pivotal role of comorbidity indices like the Age-adjusted Charlson Comorbidity Index in predicting survival outcomes is both apt and enlightening.

In response to your critique, we've taken a novel approach to reframe our manuscript. Rather than presenting our findings as a predictive model, we've chosen to pivot towards a narrative that emphasizes the discovery and elucidation of risk factors associated with early mortality in advanced endometrial cancer. This shift allows us to sidestep the limitations imposed by the lack of comorbidity data in the SEER database, focusing instead on what our data can robustly support.

Here are the innovative steps we've taken to address your concerns:

1. Narrative Shift: We've reframed our manuscript to tell a story of discovery, focusing on the identification of risk factors rather than the prediction of outcomes. This narrative shift allows us to highlight the importance of our findings without overstepping the bounds of our data.

2. Risk Factor Exploration: We've delved deeper into the identified risk factors, exploring their mechanistic roles and potential interactions. This approach not only adds depth to our analysis but also provides a foundation for future studies to build upon.

3. Limitation Embracement: Instead of viewing the absence of comorbidity data as a mere limitation, we've embraced it as an opportunity to call for a more holistic approach to data collection in future studies. We believe this proactive stance can inspire a new wave of research that addresses these gaps.

4. Vision for the Future: We've crafted a vision for future research that goes beyond the inclusion of comorbidity data. We propose a multi-faceted approach that includes not only the integration of comorbidity indices but also the exploration of genetic, lifestyle, and environmental factors that may influence survival outcomes.

5. Model as a Springboard: We position our model not as the final word but as a springboard for future research. Our hope is that it will inspire others to take up the challenge of refining and expanding upon our work, ultimately leading to more accurate and comprehensive predictive models.

By taking these innovative steps, we aim to transform the limitations of our study into a catalyst for future research. We believe that this approach not only addresses your concerns but also adds a unique and compelling dimension to our manuscript.

Thank you once again for your valuable feedback. We are excited about the potential of our revised manuscript to contribute to the field and look forward to any further insights you may have.

7. P in P-value should be italic.

Thank you for pointing out the importance of typographical accuracy in scientific writing. You are correct that the "P" in "P-value" should be italicized to distinguish it as a statistical term. This convention is important for maintaining clarity and consistency in scientific communication.

We appreciate your attention to detail and will ensure that the "P" in "P-value" is italicized throughout the manuscript to adhere to the standard typographical guidelines for statistical symbols and terms.

8. CI in Table 2 should have either – sign or comma but not Tilda sign (~).

Thank you for your careful review and for pointing out the inconsistency in the use of symbols in Table 2. You are correct that the confidence interval (CI) should be denoted with an en dash (–) or a comma, rather than a tilde (~), to maintain standard scientific notation and ensure clarity.

We apologize for the oversight and appreciate your attention to this detail. We will correct the symbols in Table 2 to use an en dash for the CI, which is the appropriate symbol for indicating a range, as per standard typographical conventions in scientific writing .

Thank you again for your valuable feedback. We will make the necessary revisions to ensure that our manuscript adheres to the highest standards of clarity and precision.

9. In line 118: “An ROC curve closer to 1 reflects a model with high TPR and low FPR, making it a better-performing model”. Sentences like this may be deleted depending on the background of the readers of this article.

Thank you for your feedback on the manuscript. You've pointed out that the sentence on line 118, which describes the interpretation of the ROC curve, may not be necessary depending on the background of the readership. This is a valid consideration, as the audience for our article may already be familiar with the basics of ROC analysis.

To address your comment, we have decided to revise the manuscript. We will remove the sentences like on line 118 to avoid redundancy and assume a level of prior knowledge among our readers that is appropriate for the journal's scope. We believe this change will tighten the narrative flow and focus the article more directly on our novel findings and contributions.

We appreciate your insight and are confident that this revision will make the article more concise and suitable for the intended audience.

Thank you again for your helpful feedback.

---

## [Decision Letter · Decision Letter 1]

30 Dec 2024

PONE-D-24-32320R1Identifying Risk Factors for Cancer-Specific Early Death in Patients with Advanced Endometrial Cancer: A Preliminary Predictive Model Based on SEER DataPLOS ONE

Dear Dr. Yang,

Thank you for submitting your manuscript to PLOS ONE. After careful consideration, we feel that it has merit but does not fully meet PLOS ONE’s publication criteria as it currently stands. Therefore, we invite you to submit a revised version of the manuscript that addresses the points raised during the review process.

**ACADEMIC EDITOR: **

The manuscript is nearly ready but requires minor corrections.

We look forward to receiving your revised manuscript.

Kind regards,

Bella Stevanny

Academic Editor

PLOS ONE

Journal Requirements:

Reviewers' comments:

Reviewer's Responses to Questions

**Comments to the Author**

1. If the authors have adequately addressed your comments raised in a previous round of review and you feel that this manuscript is now acceptable for publication, you may indicate that here to bypass the “Comments to the Author” section, enter your conflict of interest statement in the “Confidential to Editor” section, and submit your "Accept" recommendation.

Reviewer #3: All comments have been addressed

Reviewer #4: All comments have been addressed

Reviewer #5: All comments have been addressed

2. Is the manuscript technically sound, and do the data support the conclusions?

Reviewer #3: Yes

Reviewer #4: No

Reviewer #5: Yes

3. Has the statistical analysis been performed appropriately and rigorously? 

Reviewer #3: Yes

Reviewer #4: Yes

Reviewer #5: Yes

4. Have the authors made all data underlying the findings in their manuscript fully available?

Reviewer #3: Yes

Reviewer #4: Yes

Reviewer #5: Yes

5. Is the manuscript presented in an intelligible fashion and written in standard English?

Reviewer #3: Yes

Reviewer #4: Yes

Reviewer #5: Yes

6. Review Comments to the Author

Reviewer #3: Thank you for applying my comments correctly. I have two further comments:

1. In the Results section of the Abstract, active voice was used whereas passive voice was used throughout the article. It is better to be consistent.

2. First paragraph of the discussion section seems better fit in the methods section of the article; rather than a matter of discussion of results/findings.

Reviewer #4: Based on the seer database, the characteristics of diseases were analyzed in large samples to provide clinical diagnosis and treatment strategies and guide clinical treatment. However, the innovation of this paper is insufficient. There are a large number of published studies on the characteristics of endometrial cancer in the seer database. However, the conclusion does not have much significance for the clinical work.

Reviewer #5: 1. The study presents a cohort study to identify key risk factors for cancer-specific early death in advanced endometrial cancer and develop a predictive nomogram. The treatment factor depends on clinical decisions of intervention. The addition of clinical guidelines or decision pathways can support applications.

2. I am confused with the definition of early death, which might vary because the majority of deaths occur from non-cancer causes. The details of non-cancer death can improve reproducibility, such as from DVT, treatment-related or drug adverse events.

3. In inclusion criteria: The primary site of the tumor is the uterus and the first primary cancer is endometrial cancer. Were there include sarcoma?

7. PLOS authors have the option to publish the peer review history of their article (what does this mean? ). If published, this will include your full peer review and any attached files.

**Do you want your identity to be public for this peer review?** For information about this choice, including consent withdrawal, please see our Privacy Policy .

Reviewer #3: **Yes: ** Dr Shah-Jalal Sarker

Reviewer #4: No

Reviewer #5: **Yes: ** Rakchai Buhachat

---

## [Author Response · Author response to Decision Letter 2]

14 Jan 2025

Reviewer #3: Thank you for applying my comments correctly. I have two further comments:

1. In the Results section of the Abstract, active voice was used whereas passive voice was used throughout the article. It is better to be consistent.

Dear Reviewer,

Thank you for your careful review and valuable feedback on our manuscript. We greatly appreciate your observation regarding the inconsistency in voice usage between the Results section of the Abstract and the rest of the article. We agree that maintaining consistency is crucial for the clarity and coherence of the manuscript.

In response to your comment, we have revised the Abstract section to ensure that the passive voice is used consistently throughout the article. This adjustment aligns with the overall style of the manuscript and enhances its uniformity.

Once again, we sincerely thank you for your constructive suggestion, which has helped us improve the quality of our work.

2. First paragraph of the discussion section seems better fit in the methods section of the article; rather than a matter of discussion of results/findings.

Dear Reviewer,

Thank you for your careful review and constructive feedback on our manuscript. We greatly appreciate your observation that the first paragraph of the Discussion section would be more appropriately placed in the Methods section. We agree that this adjustment improves the logical flow and structure of the manuscript.

In response to your comment, we have relocated the definition of early mortality to the Methods section. Additionally, we have removed this content from the Discussion section to ensure clarity and focus on the interpretation of results.

Once again, we sincerely thank you for your valuable suggestion, which has helped us enhance the organization and quality of our work.

Reviewer #4: Based on the seer database, the characteristics of diseases were analyzed in large samples to provide clinical diagnosis and treatment strategies and guide clinical treatment. However, the innovation of this paper is insufficient. There are a large number of published studies on the characteristics of endometrial cancer in the seer database. However, the conclusion does not have much significance for the clinical work.

Thank you for your thoughtful feedback regarding the innovation and clinical significance of our study.

Regarding the innovation of our study:

We acknowledge that there are numerous published studies analyzing the characteristics of endometrial cancer based on the SEER database. However, our study focuses specifically on advanced endometrial cancer, particularly on early mortality in this population, which has been less explored in the literature. By identifying clinically accessible and relevant risk factors, our findings provide valuable insights that can aid in informed decision-making for patient management.

Regarding the clinical significance:

Our study aims to address a critical gap in the management of advanced endometrial cancer. The identified risk factors assist clinicians in the early identification of high-risk patients, enabling timely interventions and tailored treatment strategies. Furthermore, we have developed a predictive model for early mortality in advanced endometrial cancer. Given the current dearth of such prediction models, our study serves as a pioneering effort in this area. We hope that our model will provide a scientific foundation for future research and encourage other researchers to validate and build upon it for further development.

We believe that these aspects highlight the novelty and clinical relevance of our work. Once again, we sincerely thank you for your valuable suggestions, which have helped us refine and strengthen our manuscript.

Reviewer #5: 1. The study presents a cohort study to identify key risk factors for cancer-specific early death in advanced endometrial cancer and develop a predictive nomogram. The treatment factor depends on clinical decisions of intervention. The addition of clinical guidelines or decision pathways can support applications.

Thank you for your insightful comment regarding the inclusion of clinical guidelines or decision pathways to support the application of our findings.

1. Study Focus and Contribution: Our study focuses on identifying key risk factors for cancer-specific early death in advanced endometrial cancer and developing a predictive nomogram. We aimed to create a model based on accessible and clinically relevant variables to aid in early risk stratification and individualized treatment planning.

2. Clinical Decision Factors: We acknowledge that treatment decisions often depend on clinical judgment, patient preferences, and institutional practices. While the current study does not incorporate detailed clinical guidelines or decision pathways, we agree that these elements would enhance the applicability of the findings.

3. Future Research Directions: We propose that future studies should integrate clinical guidelines or decision pathways to validate and extend our nomogram’s utility. Collaborative efforts involving clinical experts and guideline developers can bridge the gap between statistical models and practical implementation in real-world settings.

We appreciate your valuable suggestion and have noted this as an important area for further exploration in the discussion section of our manuscript. Thank you again for your constructive feedback.

2. I am confused with the definition of early death, which might vary because the majority of deaths occur from non-cancer causes. The details of non-cancer death can improve reproducibility, such as from DVT, treatment-related or drug adverse events.

Thank you for your insightful comment regarding the definition of early death and the potential influence of non-cancer-related deaths. In our study, early death was defined as death occurring within six months after the initial pathological diagnosis of endometrial cancer, which is based on relevant literature and clinical practice. We acknowledge that the majority of early deaths may occur due to non-cancer causes, such as deep vein thrombosis (DVT), treatment-related complications, or adverse drug reactions.

This is indeed a limitation of our study, primarily due to the lack of detailed comorbidity data in the SEER database. To address this limitation, we have taken the following steps:

Emphasis on Cancer-Specific Early Mortality Risk Factors: We have adjusted the manuscript to place greater emphasis on identifying risk factors specifically associated with cancer-specific early death, rather than all-cause mortality. While our study does not include co-morbidities due to limitations in the available data, we have focused on the risk factors that are accessible and clinically relevant, which can aid in informed decision-making for patient management.

Acknowledgment of Limitations: We have explicitly acknowledged this limitation in the discussion section of our manuscript. We clarified that the study's predictive ability may be limited in real-world settings, where co-morbidities play a significant role in patient outcomes.

Future Research Directions: We propose that future research should aim to incorporate comorbidity data to enhance the study's predictive accuracy and applicability. This could involve multicenter, prospective studies or collaborations with other databases that contain more comprehensive patient health information.

3. In inclusion criteria: The primary site of the tumor is the uterus and the first primary cancer is endometrial cancer. Were there include sarcoma?

Thank you for pointing out the need for clarification regarding the inclusion criteria. Based on your suggestion, we have revised the manuscript to explicitly state the inclusion criteria. The updated criteria are as follows:

The inclusion site codes were C54.0 (Endometrial carcinoma of the isthmus uteri), C54.1 (Endometrial carcinoma of the corpus uteri), C54.2 (Endometrial carcinoma of the fundus uteri), C54.3 (Endometrial carcinoma of the overlapping sites of the corpus uteri), C54.8 (Endometrial carcinoma of the overlapping sites of the uterus), C54.9 (Endometrial carcinoma of the uterus, unspecified), and C55.9 (Malignant neoplasm of the uterus, unspecified). The histological codes were 8380/3 (Endometrioid adenocarcinoma), 8441/3 (Serous cystadenocarcinoma), 8480/3 (Mucinous adenocarcinoma), 8020/3 (Undifferentiated carcinoma), and 8930/3 (Adenosarcoma), according to the International Classification of Tumor Diseases, Third Edition (ICD-O-3).

Thus, endometrial stromal sarcoma (ICD-O-3 code 8930/3) was included in the study as it falls under the category of endometrial cancer with a specific histological type.

This modification ensures greater clarity and aligns with the histological types included in our study. We hope this addresses your concern. Thank you again for your valuable feedback.

---

## [Editor Report · Decision Letter 2]

21 Jan 2025

Identifying Risk Factors for Cancer-Specific Early Death in Patients with Advanced Endometrial Cancer: A Preliminary Predictive Model Based on SEER Data

PONE-D-24-32320R2

Dear Dr. yang,

We’re pleased to inform you that your manuscript has been judged scientifically suitable for publication and will be formally accepted for publication once it meets all outstanding technical requirements.

Kind regards,

Bella Stevanny

Academic Editor

PLOS ONE
---

## [Editor Report · Acceptance letter]

PONE-D-24-32320R2

PLOS ONE

Dear Dr. Yang,

I'm pleased to inform you that your manuscript has been deemed suitable for publication in PLOS ONE. Congratulations! Your manuscript is now being handed over to our production team.

Kind regards,

on behalf of

Dr. Bella Stevanny

Academic Editor

PLOS ONE